# Continuity of care in general practice in Norway

**Lamija Delalic**[1]*, **Mari Grøsland**[1], **Geir Godager**[2,3], **Henning Øien**[1,2]

**1** Department of Health Services Research, Norwegian Institute of Public Health, Oslo, Norway,
**2** Department of Health Management and Health Economics, University of Oslo, Oslo, Norway, **3** Health Services Research Unit, Akershus University Hospital, Oslo, Norway

* lamija.delalic@hotmail.com

## Abstract

### Aims

Maintaining continuity of care between doctors and patients is considered a fundamental aspect of quality in primary healthcare. In this study, we aim to examine continuity in Norway over time by computing two commonly used indicators of continuity: the *St Leonard's Index of Continuity of Care (SLICC)* and the *Usual Provider of Care Index (UPC)*.

### Method

We employ individual-level data, which covers all primary care consultations. This data includes the identities of each patient and physician, and we can identify each patient's regular general practitioner (GP). The SLICC is calculated as the share of consultations conducted by the patient's regular GP annually from 2006 to 2021. Additionally, we identify each patient's most visited physician and compute the UPC as the share of total consultations conducted by the most visited physician during the same period. Our analysis is conducted at the national level and stratified according to the level of centrality, differentiating between areas of high, moderate, and low centrality.

### Results

Our findings reveal that, at the national level, SLICC and UPC exhibit remarkable stability, reaching 64 and 71 percent, respectively, in 2021. However, there is significant geographical variation, with the least central areas experiencing less continuous healthcare (SLICC at 49 percent in 2021) than patients residing in more central areas (SLICC at 68 in 2021).

### Conclusion

Our results demonstrate a high degree of continuity that has been stable over time. However, large geographical variations suggest that policymakers should strive to reduce geographical disparities in healthcare quality.

**Data Availability Statement:** Data cannot be shared publicly because these data contain potentially identifying or sensitive patient information, and are not openly available. Researchers wishing to use this data must submit a formal application, which is accessible at https://

helsedata.no (e-mail: service@helsedata.no). This application should outline a comprehensive research proposal, detailing the project's objectives and methodologies. Approval from the Regional Committees for Medical and Health Research Ethics is also required, application for approval is available at rekportalen.no(e-mail: post@helseforskning.etikkom.no).

**Funding:** The author(s) received no specific funding for this work.

**Competing interests:** The authors have declared that no competing interests exist.

# 1 Introduction

Continuity of care stands as a fundamental pillar of patient-centered medicine [1]. Within primary care, general practitioners (GPs) hold a central role in maintaining this continuity, as they serve as the initial point of contact for both primary and specialized medical care, thus contributing significantly to early disease detection and overall well-being.

In Norway, the Regular General Practitioner (RGP) scheme entitles each patient to be listed with a specific GP. As a result, the discourse on continuity within Norway's primary care sector has predominantly revolved around examining the longevity of the patient-GP relationship [2–4] and its potential effects on patient outcomes [5]. However, the emphasis on the longevity of the patient-GP relationship does not provide a holistic view of continuity. To begin with, these studies concentrate on the duration for which a patient is registered with a particular GP, despite the fact that patients may actually seek care from different doctors within that timeframe. Furthermore, since patients can change GPs twice a year, the potential exists for patient preferences to introduce bias into the findings concerning the impact of continuity on patient outcomes.

To augment these perspectives, we aim to contribute to existing studies by examining the patient-provider relationship (coined interpersonal continuity of care [6]) by calculating two indicators: the St Leonard's Index of Continuity and Care (SLICC) and the Usual Provider of Care (UPC) Index. Previous studies have explored the continuity of interpersonal care in Norway, yet these investigations were confined to individual years or focused exclusively on a specific patient demographic. [7, 8]. We contribute to the literature by evaluating both SLICC and UPC over a longer period of time, employing exhaustive register data that covers the entire population from 2006 to 2021. It is essential to monitor continuity of care, especially in the context of ongoing challenges in retaining general practitioners (GPs) and addressing physician attrition. These issues have been of increasing concern in many countries in recent years [9–11]. In Norway, these developments have been particularly concerning in less central areas [12]. Therefore, we investigate the continuity of care at both the national level and broken down by centrality level.

# 2 Methods

We utilize nationwide individual-level data from the Norwegian Control and Distribution of Health Reimbursement database, which covers reimbursements made to all GPs following a consultation or patient contact. This data contains a patient-physician link, which allows us to observe the identity of both patient and doctor for each contact. We further link this to data from The Norwegian Health Economics Administration, containing information on each patient's regular GP at the beginning of a year, as well as the municipality in which the GP practices. Together, these data allow us to identify whether consultations were conducted by the patient's regular GP as opposed to other doctors. Data accessed for research purposes 02.11.2022. Verbal nor written consent was sought from the study population as the current investigation relied on routinely collected and anonymized register data.

To evaluate the continuity of care from 2006–2021, we compute two indicators for each calendar year (from January to December): the St Leonard's Index of Continuity and Care (SLICC) and the Usual Provider of Care (UPC) Index. While the two are sometimes used interchangeably, SLICC is most often used to measure continuity in GP practices where patients are affiliated with one specific GP [13], such as in the Norwegian context. On the other hand, the UPC index is more commonly used in countries where individuals aren't exclusively linked to a specific GP. As all Norwegians are entitled to a regular GP, but also may consult with other doctors, and to ensure that the results are comparable across various

countries and GP systems, we calculate both indices. For all years an individual has a given regular GP, we determine their usual provider of care (UPC), hereafter the most visited physician, by identifying the physician with whom the patient consulted the most out of all the patients' consultations. The most visited physician aligns with the patient's regular GP in 63 percent of all patient cases. The SLICC measures the proportion of consultations conducted by the patient's regular GP, determined by dividing the number of consultations with the regular GP in a given year by the total number of consultations with any doctor in the same year. Conversely, the UPC assesses the proportion of consultations conducted by the most visited physician, computed by dividing the number of consultations with the most visited doctor a given year by the total number of consultations with any doctor during the same year.

All in-person consultations at a (publicly funded) GP office, electronic consultations, as well as consultations at the emergency medical service are included.

We used Statistics Norway's centrality index [14], rated 1 (most central) to 6 (least central), to categorize GP offices municipalities into three groups: highly central (1–2), moderately central (3–4), and least central (5–6). Statistics Norway (2024) gives the following definition of centrality *"Centrality is a measure of a municipality's geographical position seen in relation to a centre where higher order of functions (central functions as bank, post office) is found."* The Regulation relating to a Municipal Regular General Practitioner Scheme in Norway states that municipalities shall ensure individuals residing in the municipality have access to necessary GP services [15]. This requires, among other things, the residents' GP office to be located close to the resident. Around 0.01% of residents were excluded due to missing municipality data.

## 3 Results

Over the years, the total number of patients with at least one consultation in a given year increased from 3,163,070 patients in 2006 to 3,838,052 patients in 2021. A similar trend was observed in the number of physicians conducting at least one consultation (see S1 Table in the S1 Appendix). The ratio between the number of patients with at least one consultation and the number of physicians with at least one consultation has remained relatively stable over the period, from 488 in 2006 to 422 in 2021.

### St. Leonard's Index of Continuity of Care (SLICC)

The proportion of consultations with patients' regular GP (SLICC) has been remarkably stable over time, from around 62 percent in 2006 to 64 percent in 2021 (solid line in Fig 1). This corresponds to an average yearly increase of 0.032 percentage points. This stable trend is seen at all centrality levels. However, there are relatively large level differences between the highly central and least central areas (dashed lines in Fig 1). In 2021, the proportion of consultations with patients' regular GP was 68 percent in the highly central areas, compared to only 49 percent in the least central areas.

### Usual Provider of Care (UPC) index

The proportion of consultations with the patients' most visited physician has also been stable over time (Fig 2), mirroring the consistency in SLICC (Fig 1). There is a stable trend across all centrality levels. In 2021, the nationwide UPC was 71 percent, nearly identical to UPC in 2006. However, patients in highly central areas had 74 percent of their consultations with their most visited physician, compared to 61 percent in least central regions. The increase in continuity observed in 2020 is likely caused by a substantial increase in e-consultations during the COVID-19 pandemic. According to Pedersen et al. (2022), more than 470 000 e-consultations were delivered in March 2020 whereas the corresponding number was less than 50 000 e-

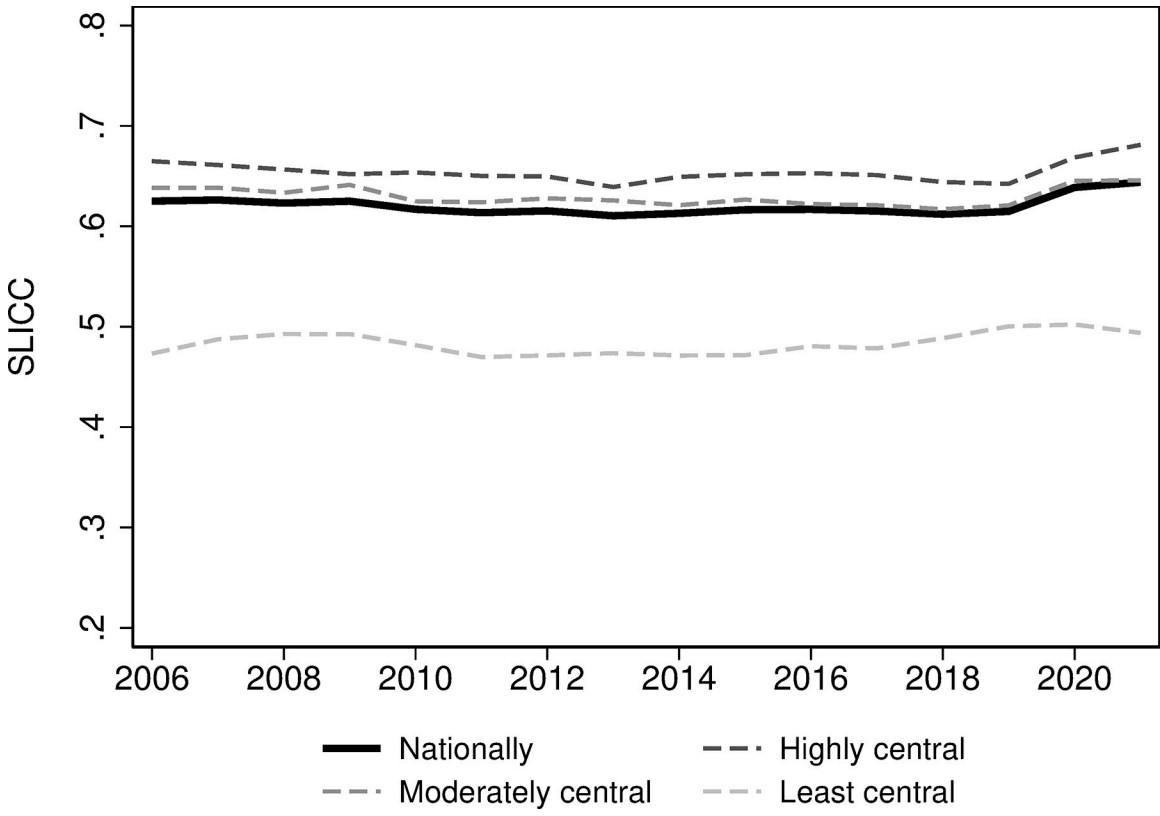

**Fig 1. St. Leonard's Index of Continuity of Care (SLICC).**

consultations during February 2020. The system for e-consultations were set up to link regular GPs and patients.

## Discussion

Utilizing individual-level data encompassing all consultations conducted during the study period, alongside information on patients' regular GPs, this study is the first to assess the development in continuity over time in Norway by utilizing two closely related indices (SLICC and UPC).

Our study reveals that the continuity of care in primary care has remained remarkably stable for the past 15 years, followed by a slight increase in recent years. The SLICC shows that between 61 and 64 percent of all consultations were performed by the patient's GP in the study period. The stable trend is also seen in the UPC index, lying at a higher level but with greater variations at 67–71 percent, suggesting that the absence of a regular GP does not necessarily negatively impact the residents' continuity in primary healthcare. Compared to other countries, our findings indicate a notably high level of continuity. For instance, a study conducted in London involving 126 practices reported a mean UPC of 0.52 in 2017–2018 [16]. However, there are notable differences in continuity between geographical areas, with rural areas experiencing lower continuity compared to more central areas. This is an important finding, calling for interventions to secure equitable continuity in rural areas of Norway as well.

A major strength of this study is that we have complete data encompassing all (publicly funded) consultations carried out in Norway over the study period, including all individuals visiting GP offices and emergency call centers. Another method for evaluating continuity is to

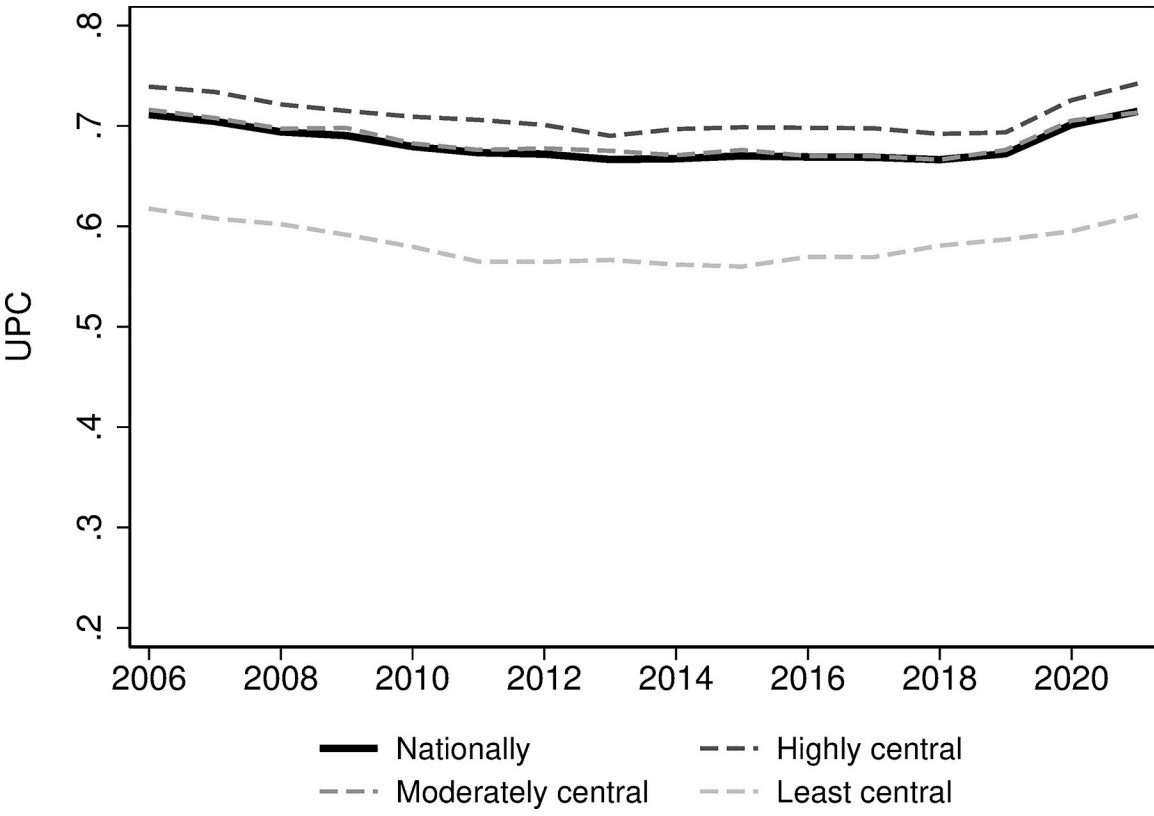

**Fig 2. Usual Provider of Care (UPC) index.**

analyze the frequency of GPs change in workplaces over time. If GPs are more frequently changing workplaces today than in previous years, it could potentially impact the continuity of care experienced by residents. One limitation of this study is, therefore, that our measure of continuity does not take such changes into account and that our analysis lays no ground for speculation on causes that might have influenced the development of continuity. Another limitation of our study is that we have not examined differences in continuity based on underlying demographic changes that might have happened over the years of study, including changes in the population or in the number of physicians practicing at GP offices. As stated in the results section, the number of residents seeking health care in general medical practice has been balanced out by an increase in the practitioners available to consult with patients. However, we have not controlled for changes in the number of consultations per patient or why they are seeking medical help. Nor have we controlled for other sociodemographic or socioeconomic variables such as gender, age, immigrant background, income, and underlying health. Future research should aim to investigate continuity disparities broken down by such characteristics.

## Conclusion

This study reveals a remarkable stability in care continuity in Norwegian general practice from 2006 to 2021. However, there are relatively large geographical disparities, with the least central areas experiencing less continuous healthcare in primary care than patients residing in more central areas. Our study of continuity offers valuable insights into the development in continuity in general practice, most importantly for policymakers aiming to reduce geographical disparities in healthcare quality.

## Supporting information

**S1 Appendix.**
(DOCX)

## Author Contributions

**Conceptualization:** Lamija Delalic, Mari Grøsland, Geir Godager, Henning Øien.

**Formal analysis:** Lamija Delalic, Mari Grøsland.

**Investigation:** Lamija Delalic.

**Methodology:** Lamija Delalic, Mari Grøsland, Geir Godager.

**Project administration:** Lamija Delalic.

**Supervision:** Geir Godager, Henning Øien.

**Writing – original draft:** Lamija Delalic, Mari Grøsland.

**Writing – review & editing:** Lamija Delalic, Mari Grøsland, Geir Godager, Henning Øien.

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
