## [Decision Letter · Decision Letter 0]

27 Mar 2024

PONE-D-24-03545Continuity of Care in General Practice in NorwayPLOS ONE

Dear Dr. Delalic,

Thank you for submitting your manuscript to PLOS ONE. After careful consideration, we feel that it has merit but does not fully meet PLOS ONE’s publication criteria as it currently stands. Therefore, we invite you to submit a revised version of the manuscript that addresses the points raised during the review process.

We look forward to receiving your revised manuscript.

Kind regards,

Rizwan Shahid, PhD

Academic Editor

PLOS ONE

2. We notice that you indicated that this is a quality improvement study and that ethical approval was not required. Please can you state this and provide details of authorities that approved this clinical audit in your ethics statement.

4. In the online submission form you indicate that your data is not available for proprietary reasons and have provided a contact point for accessing this data. Please note that your current contact point is a co-author on this manuscript. According to our Data Policy, the contact point must not be an author on the manuscript and must be an institutional contact, ideally not an individual. Please revise your data statement to a non-author institutional point of contact, such as a data access or ethics committee, and send this to us via return email. Please also include contact information for the third party organization, and please include the full citation of where the data can be found.

Reviewers' comments:

Reviewer's Responses to Questions

**Comments to the Author**

1. Is the manuscript technically sound, and do the data support the conclusions?

Reviewer #1: Yes

Reviewer #2: Yes

2. Has the statistical analysis been performed appropriately and rigorously? 

Reviewer #1: No

Reviewer #2: Yes

3. Have the authors made all data underlying the findings in their manuscript fully available?

Reviewer #1: No

Reviewer #2: Yes

4. Is the manuscript presented in an intelligible fashion and written in standard English?

Reviewer #1: Yes

Reviewer #2: Yes

5. Review Comments to the Author

Reviewer #1: The study was designed to examine the continuity of care in Norway over a 15 year period (2006-2021). The authors used two continuity of care indices, the St. Leonard’s Index of Continuity of Care (SLICC) and the Usual Provider of Care (UPC), and stratified the data by geographical level (national, high, moderate, and low centrality). They found the SLICC and UPC were relatively stable over the time period at the national level but varied widely depending on the geographical level.

Overall, I found the paper well-written, with the purpose clearly articulated. Calculating continuity of care over this time period would be valuable addition to the literature.

However, I found the methods section lacking detail in many areas. For example:

- I am assuming the continuity of care indexes were calculated each year, and their year was defined as a calendar year (January – December) versus fiscal year (April-March). This point should be made more explicit.

- The authors do not mention any exclusion or inclusion criteria applied to their data. What did the authors do about patients who only had one visit to a GP? Continuity of care indices typically need at least two visits to be calculated accurately. A single visit within a year would positively skew the indices result. Were patients who did not have a visit within a year included as well?

- The total annual number of patients and physicians included in the study should also be provided.

- Since the indices were calculated over a 15 year time period, the authors should provide some type of adjustment to account for the change in the population and the physician supply. It is probable that the indices (and patient behaviour) will be impacted by the number of physicians available as well as changes in the population, and therefore, could impact the interpretation and conclusion of the study.

- The authors state on page 4 “Typically, the most visited physicians aligns with the patient’s regular GP”, however there was no correlation calculated in the methods section and no numerical value provided in the results section. In fact, that sentence and the one following “However, then the regular GP is absent….”may be better suited to the discussion section rather than the methods section.

- On both Figure 1 & 2, the solid line is labeled as Total. Is Total the results at the National level? If that is a correct assumption, the authors need to provide that explanation, or change Total to National. Also, the y-axis is not labelled on either figure.

- The continuity of care indices for each year are displayed in Figure 1 and 2. There is no mention in the methods section or on the graphs as to whether the indices are a mean, median, or something else. I am assuming the data point on the graphs are the means for each year. If so, the standard deviation or standard error of the mean should be available.

Thank you for the opportunity to review your paper. I look forward to reading and citing your paper in the future.

Reviewer #2: Methods

There is no mention of those who might not have a primary care visit in the respective year, and if they were excluded from the study population.

It might be beneficial to give more context on how centrality is defined, for instance, distance from metro cities.

Results

Results fail to mention the drop in continuity, both SLICC and UPC, until 2019 and then the sudden increase starting 2020.

Discussion

Although, the discussion talks about the increase in continuity indices in the recent year, the authors do not speculate on what can be the reasons for this increase.

Authors suggests that were was only two percent variation in continuity since 2006, however supplementary data shows UPC doing down to 67%, indicating greater variation.

Conclusion

The conclusion seems to be from be from the perspective of the patients, i.e. comparing continuity for patients in less central areas to more central areas. However, based on the methods centrality was defined based on GP’s location. If the authors suggest using the findings to reduce geographical disparities, perhaps it is important to examine continuity from the patient’s location as well.

6. PLOS authors have the option to publish the peer review history of their article (what does this mean?). If published, this will include your full peer review and any attached files.

Reviewer #1: **Yes: **Lisa L. Cook

Reviewer #2: No

---

## [Author Response · Author response to Decision Letter 0]

15 May 2024

Dear reviewers,

Thank you for reviewing our study and for providing us with your valuable feedback and insightful suggestions. This has truly improved our article, and we are grateful for your contributions. All changes to your reviewing comments are tracked in the document, according to the line numbers provided under each review, labeled “our action”. Please see our response below.

Reviewer #1: 

The study was designed to examine the continuity of care in Norway over a 15 year period (2006-2021). The authors used two continuity of care indices, the St. Leonard’s Index of Continuity of Care (SLICC) and the Usual Provider of Care (UPC), and stratified the data by geographical level (national, high, moderate, and low centrality). They found the SLICC and UPC were relatively stable over the time period at the national level but varied widely depending on the geographical level.

Overall, I found the paper well-written, with the purpose clearly articulated. Calculating continuity of care over this time period would be valuable addition to the literature.

However, I found the methods section lacking detail in many areas. For example:

- I am assuming the continuity of care indexes were calculated each year, and their year was defined as a calendar year (January – December) versus fiscal year (April-March). This point should be made more explicit.

Our response: It is true that the years are defined as a calendar year (January-December). 

Our action: We have made a clear point of out this in the Methods section, see line 77. 

- The authors do not mention any exclusion or inclusion criteria applied to their data. What did the authors do about patients who only had one visit to a GP? Continuity of care indices typically need at least two visits to be calculated accurately. A single visit within a year would positively skew the indices result. Were patients who did not have a visit within a year included as well?

Our response: Thak you for this important feedback. We agree that more information about methods, including any exclusion or inclusion criteria, should be included. In this paper all in-person consultations and electronic consultations at a GP office and at the emergency medical office are included, including consultations for individuals with only one consultation in a given year. Our SLICC and UPC measures report on the proportion of consultation delivered by the regular GP and the most visited provider, determined by dividing the number of consultation conducted by the regular GP/most visited physician a given year by the total number of consultations with any doctor the same year. The unit of study in our paper is thus consultations, and the proportion of consultations provided by the patients’ GP and most visited physician. The reported measures characterize differences at the supply side between different areas. 

Our action: We have included the following information about our calculations of the two measures in the method section, please see line 95 to 103. 

- The total annual number of patients and physicians included in the study should also be provided.

Our response: Thank you for your comment. We agree that we should include some more information on patients and consultations. 

Our action: We have provided a table in the appendix containing details on the number of patients who had at least one consultation each year, as well as the number of doctors who conducted at least one consultation each year (see table S1 in the appendix). We have also addressed these numbers in the Results section of our article, please see line 115-120.

- Since the indices were calculated over a 15 year time period, the authors should provide some type of adjustment to account for the change in the population and the physician supply. It is probable that the indices (and patient behaviour) will be impacted by the number of physicians available as well as changes in the population, and therefore, could impact the interpretation and conclusion of the study.

Our response: Thank you for stating this. 

Our action: We have made a clear point out of this in the “Discussion” section, please see line 189-194.

- The authors state on page 4 “Typically, the most visited physicians aligns with the patient’s regular GP”, however there was no correlation calculated in the methods section and no numerical value provided in the results section. In fact, that sentence and the one following “However, then the regular GP is absent….”may be better suited to the discussion section rather than the methods section.

Our response: Thank you for pointing this out. We agree that the original text was most suitable for a discussion. However, we have opted to rephrase it slightly and follow your suggestion by incorporating a numerical value to indicate the strength of the correlation. We believe the new formulations fit better within the methodology section.

Our action: We have added a sentence stating the share of which a patient’s most visited physician aligns with the patient’s regular GP, please see line 94-95 of the Methods section.

- On both Figure 1 & 2, the solid line is labeled as Total. Is Total the results at the National level? If that is a correct assumption, the authors need to provide that explanation, or change Total to National. Also, the y-axis is not labelled on either figure.

Our response: Thanks for pointing this out. 

Our action: We have now accounted for all these suggestions, as total is changed to nationally and the y-axis in the figures are named “SLICC” and “UPC”. 

- The continuity of care indices for each year are displayed in Figure 1 and 2. There is no mention in the methods section or on the graphs as to whether the indices are a mean, median, or something else. I am assuming the data point on the graphs are the means for each year. If so, the standard deviation or standard error of the mean should be available.

Our response: Thank you for your feedback. Our SLICC and UPC measures in Figure 1 and 2 report on the proportion of consultation delivered by the regular GP and the most visited provider, determined by dividing the number of consultation conducted by the regular GP/most visited physician a given year by the total number of consultations with any doctor the same year. We agree that this should be clearer in the text. 

Our action: We have elaborated on this in the method section to describe our measures more clearly, please see line 95-101. In addition, we specify the standard deviation in table S2 and S3 in the appendix. 

Thank you for the opportunity to review your paper. I look forward to reading and citing your paper in the future.

Our response: Thank you again for your time and effort in providing such constructive feedback. We greatly appreciate your contributions.

Reviewer 2

It might be beneficial to give more context on how centrality is defined, for instance, distance from metro cities.

Our response: Statistics Norway (2024) gives the following definition of centrality 

“Centrality is a measure of a municipality's geographical position seen in relation to a centre where higher order of functions (central functions as bank, post office) is found.” Høydahl (2020) provides the most recent documentation of the centrality index for Norwegian municipalities. 

Our action: We have elaborated on the centrality measure in the Methods section, please see line 107-112.

Results

Results fail to mention the drop in continuity, both SLICC and UPC, until 2019 and then the sudden increase starting 2020.

Our response: The increase in continuity observed in 2020 is likely caused by a substantial increase in e-consultations during the covid 19 pandemic. According to Pedersen et al. (2022), more than 470 000 e-consultations were delivered in March 2020 whereas the corresponding number was less than 50 000 e-consultations during February 2020. The system for e-consultations were set up to link regular GP and patient. 

Our action: The information in our response is also included in the Results section of the manuscript, please see line 149-153.

Discussion

Although, the discussion talks about the increase in continuity indices in the recent year, the authors do not speculate on what can be the reasons for this increase.

Our response: As our aim is to contribute to the literature by providing two indices that reflect upon the continuity of care at a descriptive level, as well as to evaluate difference among centrality levels over time, our analysis lay no basis for speculations about the reasons behind this trend. However, we certainly agree that there is a need for more research to learn about the causes of this development in continuity, as well as other knowledge on how continuity may differ across different patient groups. 

Our action: We have elaborated on this point in the “Discussion” section, please see line 186-197.

Authors suggests that were was only two percent variation in continuity since 2006, however supplementary data shows UPC doing down to 67%, indicating greater variation.

Our response: Thank you for stating this mistake in the manuscript. 

Our action: We have now corrected this, please see line 162-163 of the Discussion section.

Conclusion

The conclusion seems to be from be from the perspective of the patients, i.e. comparing continuity for patients in less central areas to more central areas. However, based on the methods centrality was defined based on GP’s location. If the authors suggest using the findings to reduce geographical disparities, perhaps it is important to examine continuity from the patient’s location as well.

Our response: Thank you for your comment, adding valuable elaboration on how to understand the measure of centrality in our analysis. The Norwegian Regulation on the General Practitioner Scheme in Municipalities, section 2-3 states: "The municipality shall ensure that individuals residing in the municipality are offered necessary general practitioner services, cf. the Health and Care Services Act section 3-1 and section 3-2, first paragraph." This requires, among other things, the location of the general practitioner's office to be accessible to the municipality's residents. Therefore, the location of most of the general practitioner's office are in the municipality where the resident is residing, or by other means close to the resident. 

Our action: We have added clarification on this in the Methods section of the manuscript, in line with the existing information on the classification of centrality. Please see line 109-112.

References

Høydahl, E. (2020). Sentralitetsindeksen. Oppdatering med 2020-kommuner, Statistisk sentralbyrå. English translation: Sentrality Index. Update with 2020-municipalities.

Statistics Norway. https://www.ssb.no/en/klass/klassifikasjoner/128/om webpage accessed 25 April 2024

Pedersen K, Godager G, Rognlien HD, Tyrihjell JB, Værnø SG, Iversen T, Holte J, Abelsen B, Pahle A, Augestad L, Sæther EM (2022) Evaluering av handlingsplan for allmennlegetjenesten 2020–2024: Evalueringsrapport I. English translation: Evaluation of action plan for general praction services 2020-2024. Evaluation report I.

---

## [Decision Letter · Decision Letter 1]

27 May 2024

Continuity of care in general practice in Norway

PONE-D-24-03545R1

Dear Dr. Delalic,

We’re pleased to inform you that your manuscript has been judged scientifically suitable for publication and will be formally accepted for publication once it meets all outstanding technical requirements.

Kind regards,

Rizwan Shahid, PhD

Academic Editor

PLOS ONE

Additional Editor Comments (optional):

Reviewers' comments:

Reviewer's Responses to Questions

**Comments to the Author**

1. If the authors have adequately addressed your comments raised in a previous round of review and you feel that this manuscript is now acceptable for publication, you may indicate that here to bypass the “Comments to the Author” section, enter your conflict of interest statement in the “Confidential to Editor” section, and submit your "Accept" recommendation.

Reviewer #1: All comments have been addressed

Reviewer #2: All comments have been addressed

2. Is the manuscript technically sound, and do the data support the conclusions?

Reviewer #1: Yes

Reviewer #2: Yes

3. Has the statistical analysis been performed appropriately and rigorously? 

Reviewer #1: Yes

Reviewer #2: Yes

4. Have the authors made all data underlying the findings in their manuscript fully available?

Reviewer #1: Yes

Reviewer #2: Yes

5. Is the manuscript presented in an intelligible fashion and written in standard English?

Reviewer #1: Yes

Reviewer #2: Yes

6. Review Comments to the Author

Reviewer #1: I am satisfied with the changes the authors have made based on the review. The manuscript is well written and will be welcomed addition to the primary care continuity of care research field.

Reviewer #2: Thank you for the opportunity to review the manuscript. All the comments have been sufficiently addressed.

7. PLOS authors have the option to publish the peer review history of their article (what does this mean?). If published, this will include your full peer review and any attached files.

Reviewer #1: **Yes: **Lisa L Cook

Reviewer #2: No

---

## [Editor Report · Acceptance letter]

30 May 2024

PONE-D-24-03545R1 

PLOS ONE

Dear Dr. Delalic, 

I'm pleased to inform you that your manuscript has been deemed suitable for publication in PLOS ONE. Congratulations! Your manuscript is now being handed over to our production team.

Kind regards, 

on behalf of

Dr. Rizwan Shahid 

Academic Editor

PLOS ONE